# Dynamic Grained Encoder for Vision Transformers

**Lin Song**[1*]  **Songyang Zhang**[2,4,5*]  **Songtao Liu**[3]  **Zeming Li**[3]  **Xuming He**[2]
**Hongbin Sun**[1†]  **Jian Sun**[3]  **Nanning Zheng**[1]

[1] College of Artificial Intelligence, Xi'an Jiaotong University  [2] ShanghaiTech University
[3] Megvii Inc. (Face++)  [4]University of Chinese Academy of Sciences
[5]Shanghai Institute of Microsystem and Information Technology, Chinese Academy of Sciences
stevengrove@stu.xjtu.edu.cn, sy.zhangbuaa@gmail.com,
liusongtao@megvii.com, hexm@shanghaitech.edu.cn,
{hsun, nnzheng}@mail.xjtu.edu.cn, {lizeming, sunjian}@megvii.com

## Abstract

Transformers, the de-facto standard for language modeling, have been recently applied for vision tasks. This paper introduces sparse queries for vision transformers to exploit the intrinsic spatial redundancy of natural images and save computational costs. Specifically, we propose a Dynamic Grained Encoder for vision transformers, which can adaptively assign a suitable number of queries to each spatial region. Thus it achieves a fine-grained representation in discriminative regions while keeping high efficiency. Besides, the dynamic grained encoder is compatible with most vision transformer frameworks. Without bells and whistles, our encoder allows the state-of-the-art vision transformers to reduce computational complexity by 40%-60% while maintaining comparable performance on image classification. Extensive experiments on object detection and segmentation further demonstrate the generalizability of our approach. Code is available at https://github.com/StevenGrove/vtpack.

## 1 Introduction

Following the evolution of network architectures in natural language processing (NLP), Vision Transformers [1–5] have recently attracted increasing research attention and demonstrated promising results on several vision tasks, such as image classification, object detection, and other pixel-level tasks. Vision transformers are notable for modeling long-range dependencies and introducing less inductive bias, considered to be a solid alternative to CNNs for vision tasks.

One of the eminent obstacles for vision transformers is the high computational cost. Vision tasks typically require high-resolution image features to obtain detail and structure representation, which is critical for pixel-level tasks [6–10]. However, since the encoders in vision transformers need to establish pairwise relationships, high-resolution features could impose unacceptable computational and memory costs. Therefore, similar to the efficient transformers [11–13] in NLP, many variants [2–4] of vision transformers are proposed to perform sparse self-attentions with *dense* queries and *sparse* key-value pairs based on fixed pattern or heuristic rules.

In this paper, we notice that different from natural language, natural images involve much spatial redundancy, especially in flat or low-texture regions [14–18]. This could enable the image features to have a low resolution in some regions while maintaining similar representational capabilities.

To verify the spatial redundancy in vision transformers, we give an empirical analysis for DeiT [19] on ImageNet [20] classification dataset (the details refer to Sec. 3.1). It demonstrates the existence

---

[*]Equal contribution. This work was done in Megvii Research.
[†]Corresponding author.

35th Conference on Neural Information Processing Systems (NeurIPS 2021).

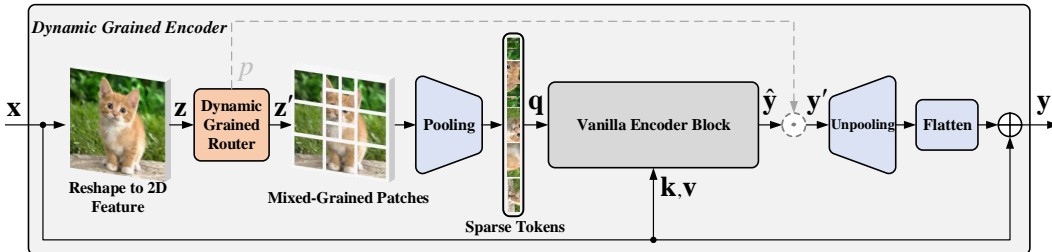

Figure 1: The overall diagram of the proposed dynamic grained encoder. $\mathbf{x}$ is the input sequence, and $\mathbf{y}$ is the output sequence. The dynamic grained router automatically split a 2D feature into mixed-grained patches with a different number of tokens in a patch. Each patch is then flattened as a sparse query by an average pooling operator. The vanilla encoder block can be a standard transformer encoder or other efficient variants. Besides, the dash lines are only used in the training phase.

of spatial redundancy in queries, and the complexity can be dramatically reduced by downsampling some highly redundant regions while maintaining comparable performance. These properties allow the queries to use mixed granularity to achieve a balance between effectiveness and efficiency, *i.e.*, more tokens in more discriminative regions while fewer tokens in less informative regions. However, the distribution of spatial redundancy varies greatly among different input images, making it difficult for a static method to handle complex and variable features.

We thus attempt to explore a new perspective: *introducing dynamic network mechanism into vision transformers to reduce the spatial redundancy of image features*. As shown in Fig. 1, we propose a Dynamic Grained Encoder (DGE) to replace the vanilla encoder in vision transformers. It could assign a suitable number of queries for each region by using a dynamic grained router, *e.g.*, the foreground regions of the cat head in Fig. 1 are assigned more queries than the background regions. Concretely, a reshaped 2D feature is first divided into regions using a fixed window. For each region, the number of patches is decided by a data-dependent routing process, and each patch is average pooled to obtain a 1D token. All the tokens are then concatenated into a sequence as the queries. Since our method focuses on the sparsity of queries, it is compatible with many efficient transformer encoders [2, 3, 11–13], making our approach available as a *generic plugin* in most vision transformers [1–3, 19, 21]. Furthermore, the output of the encoder is restored to the input resolution by an un-pooling operation and compensates for detailed information with the input feature.

To demonstrate the effectiveness, we conduct extensive experiments on three typical vision transformers, *i.e.*, DeiT [19], PVT [3] and DPVT, where DPVT is a new framework based on the deformable attention [2]. In the image classification task, our dynamic grained encoder allows these models to reduce computational complexity by 40%-60% while maintaining comparable performance. On the other hand, with lower computational complexity, the accuracy can be improved by up to 4.4% on ImageNet *val* set. In addition, the experiments on object detection and segmentation show the strong robustness and generalization of our method.

## 2 Related Work

### 2.1 Vision Transformer

Recently, Vision Transformers, inspired by the significant success of transformer [22] achieved in the NLP field, have received more attention in the vision community. ViT [1], which converts the image into a sequence and applies the transformer encoder structure directly on it for image classification, has pioneered this direction in visual recognition. To tackle the issue of training efficiency and data efficiency, DeiT [19] introduces several training strategies to enable learning the vision transformer on ImageNet. PVT [3] further develops a feature pyramid based on the transformer structure and makes it applicable for the various downstream vision tasks. Swin [21] introduces the local window idea to improve the efficiency of the transformer structure. Our work mainly focuses on reducing the spatial redundancy and improving the model efficiency in a data-dependent manner, which is rarely explored in previous works and complementary with various vision transformer structures.

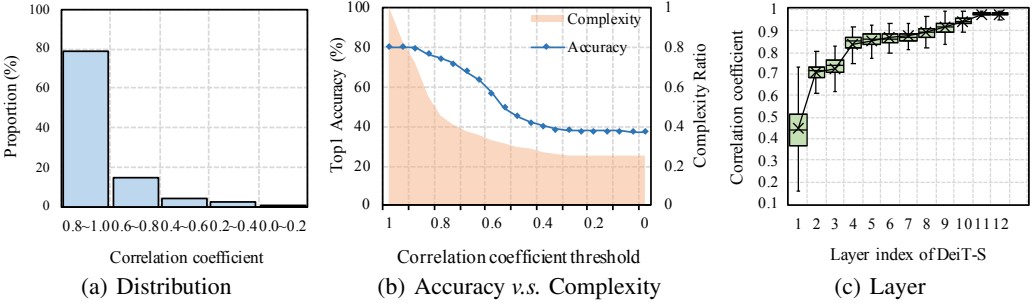

(a) Distribution       (b) Accuracy *v.s.* Complexity       (c) Layer

Figure 2: Spatial redundancy statistics of the vanilla encoders in DeiT-S [19]. The correlation coefficient is used to measure the similarity of queries in a local region. Higher the correlation corresponds to more spatial redundancy. (a) indicates that most queries are highly redundant in a local region. (b) reflects that reducing the queries with high redundancy has little impact on performance. (c) means that the redundancy varies greatly in some layers.

## 2.2 Efficient Transformer

To improve the efficiency of transformers, prior works mainly concentrate on reducing the quadratic computation of self-attention. These works can be roughly summarized as three types: learnable/fixed pattern based methods, low-rank/kernel based methods and memory based methods. Some recent approaches [12, 23–26] try to reduce the complexity of the self-attention mechanism by using a heuristic method to generate fixed or learnable patterns. Other efforts [11, 13, 27, 28] focus on utilizing the low-rank property of the attention matrix or introducing kernels to avoid computing the attention matrix explicitly. Moreover, some works [29–31] also explore the memory mechanism to improve efficiency. However, previous attempts mainly concentrate on the NLP tasks. Different from the language sequence, which has a highly abstract representation of information, natural images typically have much spatial redundancy. It makes the vision transformers require expensive costs for downstream vision tasks, especially the dense-prediction tasks, *e.g.*, object detection, segmentation. Our work tries to utilize this intrinsic property of natural images to achieve redundancy reduction in a data-dependent manner.

## 2.3 Dynamic Network

Dynamic networks [32] are proposed to adaptively change the network architecture and parameters according to input, which have been widely explored in computer vision and natural language processing tasks. Most of the dynamic networks focus on coarse-grained strategy by dropping blocks [33–36], pruning channels [37, 38] or adjusting layer-level scales [39, 40]. For instance, MSDNet [34] proposes an early existing mechanism to achieve efficient inference for image classification. Switch Transformer [41] uses the Mixture of Experts (MoE) model [42] to select different parameters for each input sample. DRNet [39] attempts to perform adaptive scale transformation in a feature pyramid network for semantic segmentation. The closest works to ours are probably the Dynamic Convolution [43] and the Dynamic Head [10], which use a learnable mask to skip specific spatial locations. However, they are only applicable to the CNN-based networks, and the skipping-location strategy could result in significant performance degradation for vision transformers (refer to Sec. 4.1.2). Different from them, our method adapts the region-level granularity to the input feature for the vision transformers, which is more general and flexible.

## 3 Method

### 3.1 Empirical Analyses on Spatial Redundancy

To investigate the spatial redundancy of vision transformer on image data, we conduct a series of experiments on the ImageNet [20] *val* set with a pre-trained DeiT-S [19] model. Our main purpose is to explore the relationship among the granularity of queries, computational complexity, and classification performance. Specifically, for each encoder layer in DeiT-S, we reshape its input

queries (excluding the extra embedding) as a 2D feature map and split it into $2 \times 2$ non-overlap patches. For each patch, we calculate its average token, and measure the similarity of each token in the patch with the average token by using the Pearson Correlation Coefficient (PCC) metric.

Then we have three valuable observations. *(1) Queries share similar patterns in a local region.* From the correlation coefficient histogram plotted in Fig. 2(a), most of the correlation coefficients are greater than 0.8, which indicates the queries typically have a strong correlation in a local region. *(2) Large potential of reducing spatial redundancy.* Furthermore, in each patch, we replace the tokens with the average token when their correlation coefficient is above a given threshold. As shown in Fig. 2(b), we illustrate the accuracy/complexity curve varying correlation thresholds. When the threshold is 0.9, the complexity decreases by 27%, but the top-1 accuracy decreases by only 0.3%. This evidence demonstrates the potential of reducing the spatial redundancy on vision transformers. *(3) Static strategy is sub-optimal.* As shown in Fig. 2(c), some encoders have large variance of correlation coefficients among different images. Thus, using data-independent methods to reduce spatial redundancy is sub-optimal, which may lead to considerable performance degradation. These observations motivate us to explore a data-dependent manner to reduce spatial redundancy.

### 3.2 Dynamic Grained Encoder

#### 3.2.1 Overall Architecture

In this paper, we propose a new encoder block for vision transformers, called *Dynamic Grained Encoder* (DGE). As shown in Fig. 1, the proposed encoder consists of two main modules, *i.e.,* dynamic grained router and vanilla encoder block. Specifically, the dynamic grained router adaptively generates mixed-grained patches for a 2D feature. The vanilla encoder block can be a standard encoder block [22] or other efficient variants [2, 9, 11–13, 44], which is made up of a multi-head attention and a feed-forward network. If there are extra tokens in the input sequence, such as class embedding in ViT [1], we handle them separately with the vanilla encoder. For ease of presentation, the rest of this section only considers the input sequence without extra tokens.

Given an input sequence $\mathbf{x} \in \mathbb{R}^{(H \times W) \times C}$ for the dynamic grained encoder, $(H, W)$ denotes the resolution of the feature, $C$ is the number of channels. To compatible with most vanilla encoders, we only generate sparse queries $\mathbf{q} \in \mathbb{R}^{N \times C}$ by the dynamic grained router, where $N$ indicates the number of queries. Then the sparse queries as well as dense keys $\mathbf{k}$ and values $\mathbf{v}$ are transformed by a vanilla encoder. It is worth mentioning that keys and values can be sparse in the vanilla encoder to improve efficiency further. The output sequence of the vanilla encoder is restored to a 2D feature with the original resolution by using an un-pooling operation. Furthermore, to enhance the details of the output feature and alleviate the vanishing gradient problem, we add a residual connection [45] to fuse the input sequence.

#### 3.2.2 Dynamic Grained Router

To achieve dynamic grained patches in space, we first partition the 2D feature, denoting as $\mathbf{z}$, into multiple regions, which can perform in regular or irregular ways. Although the irregular ways, *e.g.,* superpixels [46] and segmentation [47], may lead to better performance, it is very unfriendly to memory access and inducing inefficiency. Therefore, as shown in Fig. 3, we adopt a $S \times S$ non-overlap window[3] to split image features into multiple regular regions. Furthermore, we define a set of candidate granularities $\Phi = \{\phi_1, \phi_2, ..., \phi_K\}$ to represent the optional patch size in a region, where $K$ is the number of candidate granularities. The granularity denotes the side length of a patch, *e.g.,* $\phi = 8$ corresponds to an $8 \times 8$ patch. Since each patch is pooled into one query in the encoder, larger granularity indicates fewer queries and less computation. For convenience, we set the region size with the maximum granularity, *i.e.,* $S = \max(\Phi)$, in the experiments.

**Inference.** For a region $i \in \{1, 2, ..., \lceil \frac{H}{S} \rceil \cdot \lceil \frac{W}{S} \rceil\}$, we use a gating network to select a granularity from the set of candidate granularities. Concretely, we reduce the region feature $\mathbf{z}_i$ into a representative token by using the average pooling operation and linearly project it to the gating logits:

$$h(\mathbf{z}_i) = \frac{1}{S^2} \sum_{j=1}^{S^2} \mathbf{z}_{i,j} \mathbf{W} + b, \tag{1}$$

---

[3]Bottom-right padding is adopted on the feature if needed.

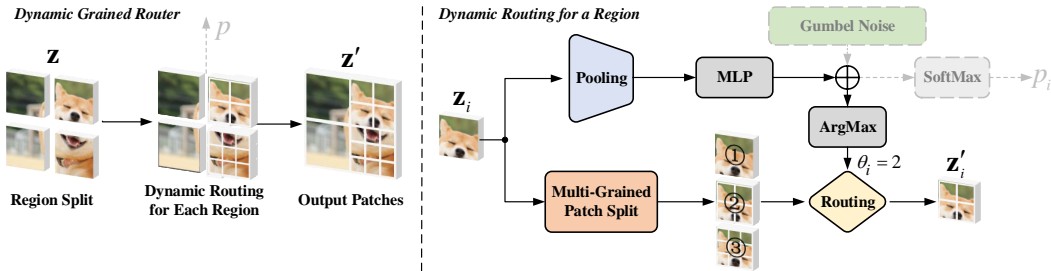

Figure 3: The diagram of the dynamic grained router in a DGE. As shown in the left part, a 2D feature is split into multiple regions. For each region, as shown in the right part, we generate multiple groups of patches with different granularities and select a specific group by the gating network. The Gumbel noise is added to achieve end-to-end training. Besides, the modules in dash lines are only used in the training phase.

where $\mathbf{W} \in \mathbb{R}^{C \times K}$ and $b \in \mathbb{R}^{1 \times K}$ indicate the weight and bias, respectively. The gating logits is used to decide the granularity for the region by calculating the gating indices:

$$\theta_i = \arg \max_k (h(\mathbf{z}_i)_k) \in \{1, 2, ..., K\}. \tag{2}$$

As shown in Fig. 3, we split the region feature into multiple groups of patches[1] with $K$ granularities. We then choose a group of specific granularity according to the gating indices. We denote the selected group as $\mathbf{z}'_i \in \mathbb{R}^{N_i \times \phi_{\theta_i}^2 \times C}$, where $N_i = \lceil \frac{S}{\phi_{\theta_i}} \rceil \cdot \lceil \frac{S}{\phi_{\theta_i}} \rceil$ is the number of patches in the group.

As shown in Fig. 1, to construct a sequence as queries, we use the spatial mean vector of each patch as the representative token by a pooling operation and concatenate all the tokens for the vanilla encoder:

$$\hat{\mathbf{y}}_i = \text{VanillaEncoder}(\mathbf{q}_i, \mathbf{k}, \mathbf{v}), \text{ where } \mathbf{q}_i = \frac{1}{\phi_{\theta_i}^2} \sum_{j=1}^{\phi_{\theta_i}^2} \mathbf{z}'_{i,j} \in \mathbb{R}^{N_i \times C}. \tag{3}$$

Compared with the previous encoders [2, 11–13], the number of queries is reduced to $1/\phi_{\theta_i}^2$ of the original, the efficiency of the encoder can be improved, and the acceleration is more significant when selected granularity $\theta_i$ is larger.

**Training.** To enable the end-to-end training for the gating network, motivated by [43, 48–50], we replace the determined decisions in Eq. 2 with a stochastic sampling process during the training phase. Specifically, given a categorical distribution with unnormalized log probabilities, a discrete gating index can be yielded with noise samples $g_j$ drawn from a standard Gumbel distribution:

$$\theta_i = \arg \max_k (h(\mathbf{z}_i)_k + g_k), \text{ where } g_k \sim \text{Gumbel}(0, 1). \tag{4}$$

Furthermore, since the Eq. 4 is a hard decision process, it is not straightforward to train the gating logits. To enable the back-propagation, we adopt the Gumbel-Softmax technique [51] to give a continuous and differentiable approximation by replacing the argmax with a softmax operation. The soft gating score for a region is then selected by the gating index:

$$p_i = \frac{\exp((h(\mathbf{x}_i)_{\theta_i} + g_{\theta_i})/\tau)}{\sum_k^K \exp((h(\mathbf{x}_i)_k + g_k)/\tau)} \in [0, 1], \tag{5}$$

where a fixed temperature $\tau = 1$ is used in our experiments for convenience. Similar with [43, 52], we further use a straight-through estimator for the gradients of gating logits, which are obtained through the soft gating score $p_i$ during the backward pass:

$$\mathbf{y'_i} = \begin{cases} \hat{\mathbf{y}} & \text{forward} \\ p_i \cdot \hat{\mathbf{y}} & \text{backward} \end{cases} \tag{6}$$

The above stochastic process is only adopted in the training phase. Our method requires no random sampling and exponential functions during inference, guaranteeing high efficiency in practice.

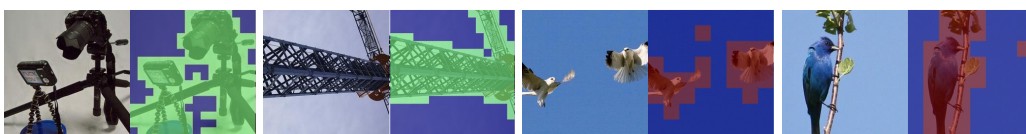

(a) gating indices of different images

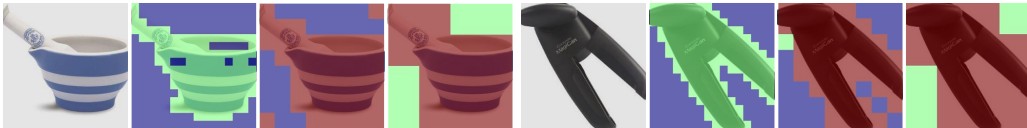

(b) gating indices of different stages

Figure 4: Visualization of predicted gating indices of PVT-S+DGE on ImageNet *val* set. The candidate granularity set is $\Phi = \{1, 2, 4\}$, which are shown in red, green and blue respectively. Higher granularity corresponds to less computational complexity. Our dynamic encoder tends to assign more queries to the representative foreground regions than the background regions, thus significantly reducing the computational cost. The left and right parts of Fig.4(a) come from stage 1 and stage 2 of PVT, respectively. From left to right, the heatmaps of each instance in Fig.4(b) correspond to stage 1, stage 2, and stage 3, respectively.

### 3.2.3 Budget Constraint

In the absence of a budget constraint, our encoder typically prefers to assign more queries to each region to achieve high performance. To obtain a better balance between effectiveness and efficiency, we define a *computational budget* denoted as $\gamma \in [0, 1]$, which corresponds to the desired computational complexity ratio relative to the vanilla encoder without dynamic grained.

Given a vision transformer with $L$ dynamic grained encoders, we can calculate the used computational complexity ratio of the transformer by:

$$\beta = \frac{\sum_l^L \mathcal{C}^l \psi^l}{\sum_l^L \mathcal{C}^l H^l W^l}, \text{ where } \psi^l = \left\{ \begin{array}{ll} \sum_i \phi_{\theta_i}^2 & \text{forward} \\ \sum_i p_i^l \cdot \phi_{\theta_i}^2 & \text{backward} \end{array} \right. \tag{7}$$

The $\mathcal{C}^l$ indicates the computational complexity required to compute a query in an encoder layer. The $\psi^l$ corresponds to the number of queries, adopting a straight-through estimator to enable end-to-end training. This strategy ensures an accurate complexity estimation when computing the training loss. Moreover, we use the Euclidean distance for the budget loss to narrow the computational complexity to a predetermined bound:

$$\mathcal{L} = \mathcal{L}_{\text{task}} + \lambda \mathcal{L}_{\text{budget}}, \text{ where } \mathcal{L}_{\text{budget}} = (\beta - \gamma)^2. \tag{8}$$

The hyper-parameter $\lambda$ balances losses among different tasks, making the gradients have the same order of magnitude. Besides, for batched image inputs, $\beta$ is averaged along the batch dimension to estimate the average load of the network.

## 4   Experiment

In this section, we apply our encoder to the state-of-the-art vision transformers and conduct extensive experiments on image classification, object detection, and segmentation. To demonstrate the generalization of our method, we conduct experiments on three Vision Transformer frameworks, *i.e.*, DeiT [19], PVT [3] and DPVT. Where DPVT is a new framework we proposed, which is based on the architecture of PVT [3] but using the deformable attention [2] as the vanilla encoder. Different from the dense self-attention process in DeiT, PVT and DPVT utilize sparse key-value pairs in position-insensitive and position-sensitive ways, respectively. These three frameworks could represent the vanilla encoder used by most vision transformers.

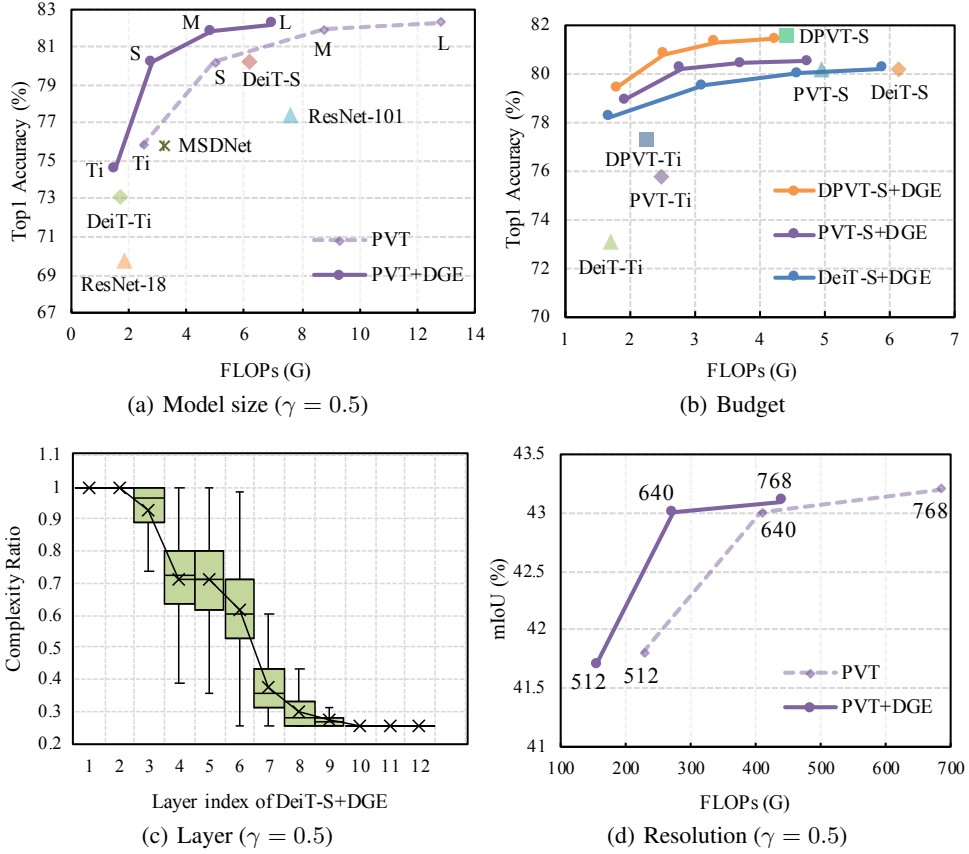

Figure 5: Visualization of accuracy and computational complexity of different configurations. (a), (b) and (c) are evaluated on ImageNet *val* set. The PVT and PVT+DGE in (a) is scaled by model size, *i.e.*, "tiny", "small" "medium" and "large". (b) indicates the performance of our method with different budget constraints. (c) reflects the distribution of computational complexity in different encoder layers of the DeiT-S+DGE. (d) is evaluated on ADE-20K *val* set with varying image resolutions.

## 4.1 Image Classification on ImageNet

### 4.1.1 Implementation Detail

All the experiments for image classification are based on ImageNet [20] classification dataset. We use $256 \times 256^4$ as the input image resolution for training and evaluation. For a fair comparison, we follow the training settings in DeiT and PVT. Specifically, the random-size cropping, random horizontal flipping [53] and mixup [54] are used for data augmentation. We use the AdamW [55] optimizer with the weight decay of 0.05 and the momentum of 0.9. The learning rate is initially set to 0.001 and decreases according to the cosine schedule [56]. All the models are trained for 300 epochs with 128 images per batch. The label-smoothing regularization is used in the training phase. Besides, for the dynamic grained encoders, $\lambda$ is set to 1.0 and $\Phi$ is set to $\{1, 2, 4\}$ by default. During the training phase, we use four compute nodes with 32 Nvidia Tesla V100 GPUs. For instance, we spend about 1.2 days training the PVT-S with DGE model for 300 epochs. For the runtime evaluation, we measure the frameworks on both Intel Xeon Gold 6130 CPU and Nvidia Tesla V100 GPU to demonstrate the efficiency of our dynamic networks.

---

[4]To achieve efficient region splitting, we choose $256 \times 256$ instead of $224 \times 224$ as it is divisible by more optional granularities. We re-train all involved vision transformers in this work for a fair comparison.

Table 1: Performance of dynamic grained encoder with different configurations on ImageNet *val* set. The budget for DGE is set to 0.5. "Region" means using region-wise routing instead of layer-wise routing in the encoder.

| Framework | Dynamic | Region | $\Phi$ | Top1 Acc | Top5 Acc | FLOPs | #Param |
|---|---|---|---|---|---|---|---|
| PVT-S | ✗ | - | - | 80.2 | 95.2 | 6.2G | 28.2M |
| PVT-S+DGE | ✓ | ✗ | 1, 2, 4 | 79.1 | 94.5 | 3.4G | +12.1K |
|  |  | ✓ | 0, 1 | 78.8 | 94.4 | 3.5G | +8.1K |
|  |  |  | 1, 2 | 80.0 | 95.0 | 3.5G | +8.1K |
|  |  |  | 1, 2, 4 | 80.2 | 95.0 | 3.5G | +12.1K |
|  |  |  | 1, 2, 4, 8 | 79.9 | 95.0 | 3.4G | +16.1K |

### 4.1.2  Ablation Study

**Where are Fine-Grained Queries Assigned?**   To reveal the undergoing properties of our dynamic grained encoder, we illustrate the predicted gating indices $\theta$ on ImageNet *val* set, which is shown in Fig. 4. Without additional supervision other than classification, our dynamic network can generate instance-aware masks with rich details. It allows the encoder to assign more queries on the foreground regions with discriminative features than background regions. This ensures that the network can consume less computational cost while maintaining fine-grained representation. In addition, as presented in Fig. 4(b), the predicted gating indices have similar patterns among different stages in the PVT. It demonstrates the effectiveness for a pyramid network, which is crucial for applying to the downstream tasks.

**Dynamic vs Static**   To demonstrate the superiority of the dynamic mechanism, we give a comparison on the PVT framework with different model sizes in Fig. 5(a). For convenience, we fix the budget constraint $\gamma$ at 0.5. Our dynamic grained encoder can reduce the computational complexity by half while maintaining comparable performance. On the other hand, with similar computational complexity, our method can improve the static transformers by up to 4.4%. The results demonstrate the effectiveness of our method even on the efficient vision transformers. In addition, as shown in Fig. 5(c), we calculate the complexity ratio of each layer in DeiT-S with DGE, where the complexity of the network in the middle layers varies significantly due to the dynamic mechanism. Interestingly, the deeper layer has lower average computational complexity, which means the deeper layer tends to assign fewer queries. Thus, *DeiT is turned into a dynamic feature pyramid structure, which is consistent with the observation in CNNs.*

**Budget Constraint and Candidate Granularity Set**   As illustrated in Fig. 5(b), we give a comparison of varying the budget constraints $\gamma$, which is selected from $\{0.25, 0.5, 0.75, 1.0\}$ respectively. The redundancy in space allows the network to achieve comparable performance with much less computational cost even on the efficient transformers, *e.g.*, PVT and DPVT. Our encoder achieves the optimal balance between effectiveness and efficiency when the budget is about half. Therefore, we set the budget constraint to 0.5 for other experiments by default. In addition, we report the performance of PVT-S with DGE with different candidate granularity set $\Phi$ in Tab. 1. When $\Phi = \{0, 1\}$, the gating indices degenerate into a learnable binary mask similar to dynamic convolutions [10, 43], but this strategy results in significant performance degradation. There is no significant difference in performance between other granularity settings. The performance is highest when $\Phi = \{1, 2, 4\}$, which becomes our default setting.

**Region-wise Routing vs Layer-wise Routing**   The Fig. 4 clearly demonstrates that DGE can perform dynamic granularity in space to adapt to different object structures. Nevertheless, most previous dynamic networks are based on layer-wise routing [32]. To demonstrate the advantages of our method, we set the region size $S \times S$ to the input feature size so that DGE can be degraded from region-wise routing to layer-wise routing. As shown in Tab. 1, region-wise gating achieves 1.1% absolute gains over layer-wise gating with similar complexity, which agrees well with the empirical analysis in Sec.3.1.

Table 2: Performance of dynamic grained encoder on COCO *val* set. All experiments are conducted with 1x schedule [57]. Time and FLOPs are measured on an $800 \times 1280$ image. "C" and "G" indicate the backbone latency on CPU (Xeon 6130) and GPU (Tesla V100). All the budget for DGE is 0.5.

| Backbone | Size | #Param (M) | Latency C(ms) | Latency G(ms) | FLOPS (G) | $AP_b$ | $AP_b^{50}$ | $AP_b^{75}$ | $AP_m$ | $AP_m^{50}$ | $AP_m^{75}$ |
|---|---|---|---|---|---|---|---|---|---|---|---|
| ResNet | 50 | 44.2 | - | - | 189 | 38.0 | 59.6 | 41.4 | 34.4 | 55.1 | 36.7 |
| PVT | Small | 44.3 | 880 | 33 | 251 | 40.4 | 62.9 | 43.8 | 37.8 | 60.1 | 40.3 |
| **PVT+DGE** | Small | 44.3 | 440 | 26 | 185 | 40.1 | 62.6 | 43.2 | 37.5 | 59.7 | 40.0 |
| DPVT | Small | 37.7 | 1090 | 50 | 186 | 44.0 | 65.9 | 48.2 | 40.3 | 62.9 | 43.4 |
| **DPVT+DGE** | Small | 37.7 | 720 | 34 | 147 | 43.8 | 65.7 | 47.7 | 40.0 | 62.6 | 43.2 |
| ResNet | 101 | 63.2 | - | - | 263 | 40.4 | 61.1 | 44.2 | 36.4 | 57.7 | 38.8 |
| ResNeXt | 101(32x4) | 62.8 | - | - | 354 | 41.9 | 62.5 | 45.9 | 37.5 | 59.4 | 40.2 |
| PVT | Medium | 63.9 | 1260 | 73 | 339 | 42.0 | 64.4 | 45.6 | 39.0 | 61.6 | 42.1 |
| **PVT+DGE** | Medium | 63.9 | 620 | 40 | 228 | 41.7 | 64.1 | 45.0 | 38.3 | 62.0 | 40.6 |
| DPVT | Medium | 49.9 | 1800 | 75 | 236 | 46.4 | 68.0 | 51.1 | 42.0 | 65.2 | 45.2 |
| **DPVT+DGE** | Medium | 49.9 | 1240 | 50 | 169 | 45.8 | 67.2 | 50.0 | 41.4 | 64.5 | 44.6 |

Table 3: Performance of different backbones for semantic segmentation on ADE-20K *val* set. The inference time (backbone) is measured for a $512 \times 2048$ input image. "C" and "G" indicate the latency on CPU and GPU.

| Backbone | #Param (M) | FLOPs (G) | mIoU (%) | Latency C(ms) | Latency G(ms) |
|---|---|---|---|---|---|
| PVT-S | 28.2 | 226 | 41.8 | 1350 | 65 |
| **PVT-S+DGE** | 28.2 | 155 | 41.7 | 720 | 42 |
| PVT-M | 48.0 | 316 | 44.0 | 1910 | 100 |
| **PVT-M+DGE** | 48.0 | 202 | 43.9 | 1100 | 64 |
| DPVT-S | 21.7 | 157 | 44.4 | 1470 | 55 |
| **DPVT-S+DGE** | 21.7 | 121 | 44.4 | 860 | 32 |
| DPVT-M | 34.3 | 209 | 46.8 | 1990 | 110 |
| **DPVT-M+DGE** | 34.3 | 148 | 46.1 | 1260 | 50 |

Table 4: Comparisons with state-of-the-art vision transformers on ADE-20K *val* set. FLOPs is tested on $512 \times 2048$ resolution.

| Backbone | #Param (M) | FLOPs (G) | mIoU (%) |
|---|---|---|---|
| ResNet-50 [45] | 28.5 | 184 | 36.7 |
| PVT-S [3] | 28.2 | 226 | 41.8 |
| Swin-Ti [21] | 31.9 | 187 | 41.5 |
| Twins-S [60] | 28.3 | 174 | 43.2 |
| **DPVT-S+DGE** | **21.7** | **121** | **44.4** |
| ResNet-101 [45] | 47.5 | 262 | 38.8 |
| PVT-M [3] | 48.0 | 316 | 44.0 |
| Swin-S [21] | 53.2 | 280 | 44.9 |
| Twins-B [60] | 60.4 | 318 | 45.3 |
| **DPVT-M+DGE** | **34.3** | **148** | **46.1** |

## 4.2 Experiments for Downstream Tasks

### 4.2.1 Object Detection/Instance Segmentation on COCO

We apply our models for object detection and instance segmentation on the COCO dataset [58]. We resize the images so that the shorter side is 768 pixels. All experiments are conducted on 8 GPUs with 2 images per GPU (effective minibatch size 16) for 90K iterations. The learning rate is initialized to 1e-4, which is decreased by 10 at the 60K and 80K iteration. Following the settings in PVT [3], we report the performance with 1x training schedule [57, 59].

The results are reported in Tab. 2. When equipped with DGE, the PVT-S achieves comparable performance at 40.1% $AP_{box}$ with a significant complexity reduction (185G vs 251G) and inference speed up by 22%. Even with larger models or different vanilla encoders, our method is still effective and efficient. In addition, the proposed vision transformer variant, *i.e.*, DPVT, is also competitive in terms of parameter, computational cost and performance. Moreover, DPVT-M+DGE achieves 45.8 $AP_{box}$ with 169G FLOPs, even efficient than the ResNet-50 backbone.

### 4.2.2 Semantic Segmentation on ADE-20K

We further evaluate our models as the backbones for Semantic-FPN [61] on ADE-20K [62] dataset. All the experiments are based on MM-Segmentation toolkit [63]. In the training phase, we follow the settings in PVT [3] and set the learning rate to 1e-4, which gradually decreases to 0 by the poly strategy [64]. The images are cropped to $512 \times 512$ and augmented with random scaling (from 0.5 to 2.0) and flipping. All models are trained in 80k iterations with a batch size of 32.

We conduct several ablation studies by introducing the DGE block into PVT [3] and our proposed DPVT. As shown in Tab. 3, with our dynamic grained encoder, DPVT+DGE and PVT+DGE both achieve competitive performance with a significant computation cost reduction by about 30% FLOPs. On the other hand, PVT-M+DGE achieves 2.1% mIoU absolute gains over PVT-S but with less computational complexity. As illustrated in Fig. 5(d), this phenomenon also occurs for different image sizes on the same framework, *e.g.*, our method has up to 1.2% mIoU absolute gains against the baseline with similar computational complexity. In addition, as shown in Tab. 4, our DPVT models with DGE are superior to the state-of-the-art vision transformers in terms of parameters, computational complexity and performance. These results well demonstrate the generalization ability and robustness of our method.

## 5  Conclusion

In this paper, we analyze the spatial redundancy in vision transformers and propose a dynamic grained encoder to speed up inference. Our encoder can adaptively yield a suitable number of queries for different regions to reduce spatial redundancy while maintaining comparable performance. Besides, our encoder is compatible with many efficient transformers and can be trained in an end-to-end manner. The extensive experiments demonstrate the effectiveness and generalization of our method. In general, this paper explores a new perspective, *i.e.*, leveraging the intrinsic properties of natural images with the dynamic network mechanism to achieve efficient vision transformers. We hope that our dynamic grained encoder can provide insights into future works and beyond.

## Acknowledgments and Disclosure of Funding

This research was supported by National Key R&D Program of China (No. 2017YFA0700800), National Natural Science Foundation of China (No. 61790563 and 61774125), Shanghai Science and Technology Program (No. 21010502700).

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
