# Dynamic Grained Encoder for Vision Transformers

## A  Limitation and Future Work

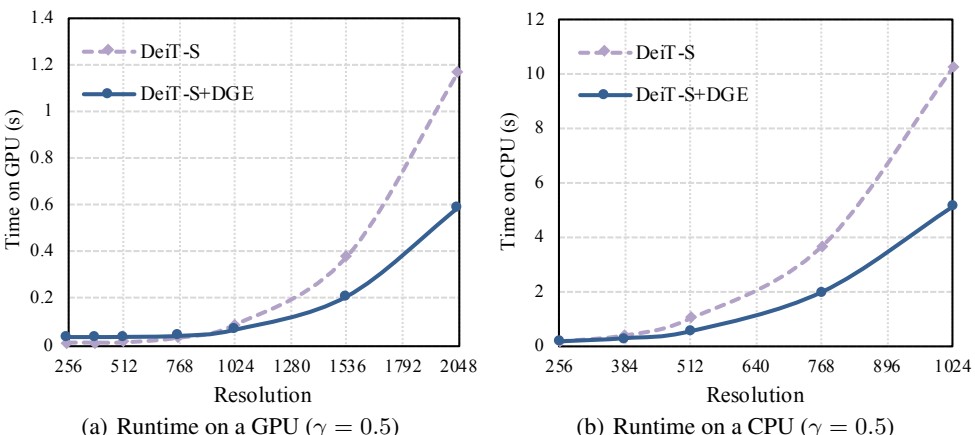

(a) Runtime on a GPU ($\gamma = 0.5$)   (b) Runtime on a CPU ($\gamma = 0.5$)

Figure 6: The comparison of inference time on a Nvidia Tesla V100 GPU or a Intel Xeon Gold 6130 CPU. The budget for DGE is set to 0.5. "Resolution" refers to the side length of input images.

As shown in Fig. 6(a), one limitation of our work is that the acceleration ratio on GPUs (based on native PyTorch implementation) is not good when the input image size is small. We suspect that this is due to the additional modules of DGE resulting in more scheduling processes, and scheduling processes lead to static time consumption. Nevertheless, our work demonstrates the superiority of efficiency on large-size input images, which is crucial for many downstream tasks and practical scenes. As illustrated in Fig. 6(b), our method also has a significant speed gain on CPUs even for small input images, making it applicable to mobile devices. We look forward to reducing the static time consumption of DGE through device-specific optimizations in future work.

## B  Additional Experiments

### B.1  Quantitative Analysis on Dynamic Grained Router

We follow the weakly supervised segmentation [64] to show how well the dense query region captures the foreground region. The metric in [64] is used to measure the gating scores in each DGE layer. Specifically, we set the candidate granularities $\Phi$ to $\{1, 2\}$, so that the finer-grained gating scores are taken as a soft-segmentation of the image. We adopt the evaluation protocol in [64] to report the quantitative segmentation results. As shown in Tab. 5 and Tab. 6, our gating scores have significant superiority even over the weakly supervised method, *i.e.*, GradCAM. These results demonstrate that the DGE could guide the transformer to focus on the foreground regions, which is consistent with the visualization.

35th Conference on Neural Information Processing Systems (NeurIPS 2021), Sydney, Australia.

Table 6: The quantitative analysis on PVT-S with DGE ($\gamma = 0.5$).

| Metric | Random | Layer 1 | Layer 6 | Layer 11 | Layer 16 |
|--------|--------|---------|---------|----------|----------|
| Accuracy | 50.0 | 55.4 | 49.1 | **67.8** | 65.5 |
| mAP | 50.0 | 68.0 | 45.2 | 71.3 | **79.4** |
| mIoU | 31.9 | 34.5 | 32.5 | **50.2** | 46.6 |

Table 5: The quantitative analysis on DeiT-S with DGE ($\gamma = 0.5$).

| Metric | Random | GradCAM [64] | Layer 1 | Layer 4 | Layer 8 |
|--------|--------|--------------|---------|---------|---------|
| Accuracy | 50.0 | 64.4 | 55.4 | 56.3 | **67.6** |
| mAP | 50.0 | 71.6 | 63.5 | 60.7 | **78.8** |
| mIoU | 31.9 | 40.8 | 36.4 | 37.7 | **48.2** |

## B.2 Runtime Analysis on GPUs

The efficiency of our DGE modules on GPUs mainly relies on the throughput of sparse matrix multiplication, which is dependent on hardware architecture and code optimization. To demonstrate the potential of our method for parallel devices, we implement an optimized CUDA kernel with multiple streams for batched sparse matrix multiplication. With this kernel, we report the runtime comparison of different backbones for multiple downstream tasks on a Tesla V100 GPU. The results are reported in Tab. 7 and Tab. 8, where the latency indicates the runtime of backbone.

Table 7: Runtime comparison of MaskRCNN (1x) framework on COCO *val* set ($\gamma = 0.5$).

| Backbone | $AP_b$ | $AP_m$ | FLOPs | Latency (CPU) | Latency (GPU) |
|----------|--------|--------|-------|---------------|---------------|
| PVT-S | 40.4 | 37.8 | 251G | 0.88s | 33ms |
| **PVT-S+DGE** | 40.1 | 37.5 | 185G | 0.44s | 26ms |
| DPVT-S | 44.0 | 40.3 | 186G | 1.09s | 50ms |
| **DPVT-S+DGE** | 43.8 | 40.0 | 147G | 0.72s | 34ms |
| PVT-M | 42.0 | 39.0 | 339G | 1.26s | 73ms |
| **PVT-M+DGE** | 41.7 | 38.3 | 228G | 0.62s | 40ms |
| DPVT-M | 46.4 | 42.0 | 236G | 1.80s | 75ms |
| **DPVT-M+DGE** | 45.8 | 41.4 | 169G | 1.24s | 50ms |

Table 8: Runtime comparison of Semantic-FPN framework on ADE20K *val* set ($\gamma = 0.5$).

| Backbone | mIoU | FLOPs | Latency (CPU) | Latency (GPU) |
|----------|------|-------|---------------|---------------|
| PVT-S | 41.8 | 226G | 1.35s | 65ms |
| **PVT-S+DGE** | 41.7 | 155G | 0.72s | 42ms |
| DPVT-S | 44.4 | 157G | 1.47s | 55ms |
| **DPVT-S+DGE** | 44.4 | 121G | 0.86s | 32ms |
| PVT-M | 44.0 | 316G | 1.91s | 100ms |
| **PVT-M+DGE** | 43.9 | 202G | 1.10s | 64ms |
| DPVT-M | 46.8 | 209G | 1.99s | 110ms |
| **DPVT-M+DGE** | 46.1 | 148G | 1.26s | 50ms |

## B.3 Implementation Details for Complexity Computation

We report the FLOPs following the conventional protocol of dynamic networks [32]. Specifically, we split the entire network into static and dynamic parts. The complexity of the static part, *i.e.*, the modules without dynamic mechanism including the gating networks in DGE, is computed in the standard way [1,3,19]. For the complexity of the dynamic part, *i.e.*, the dynamic modules in DGE, we accumulate the complexity associate with each enabled query according to the gating indices.

# C  Visualization

We provide the visualization of predicted results for object detection and instance segmentation on COCO *val* set, which is shown in Fig. 7. The visualization for semantic segmentation on ADE-20K *val* set is illustrated in Fig. 8. With similar computational complexity, our approach has advantages in terms of modeling context and structure preservation.

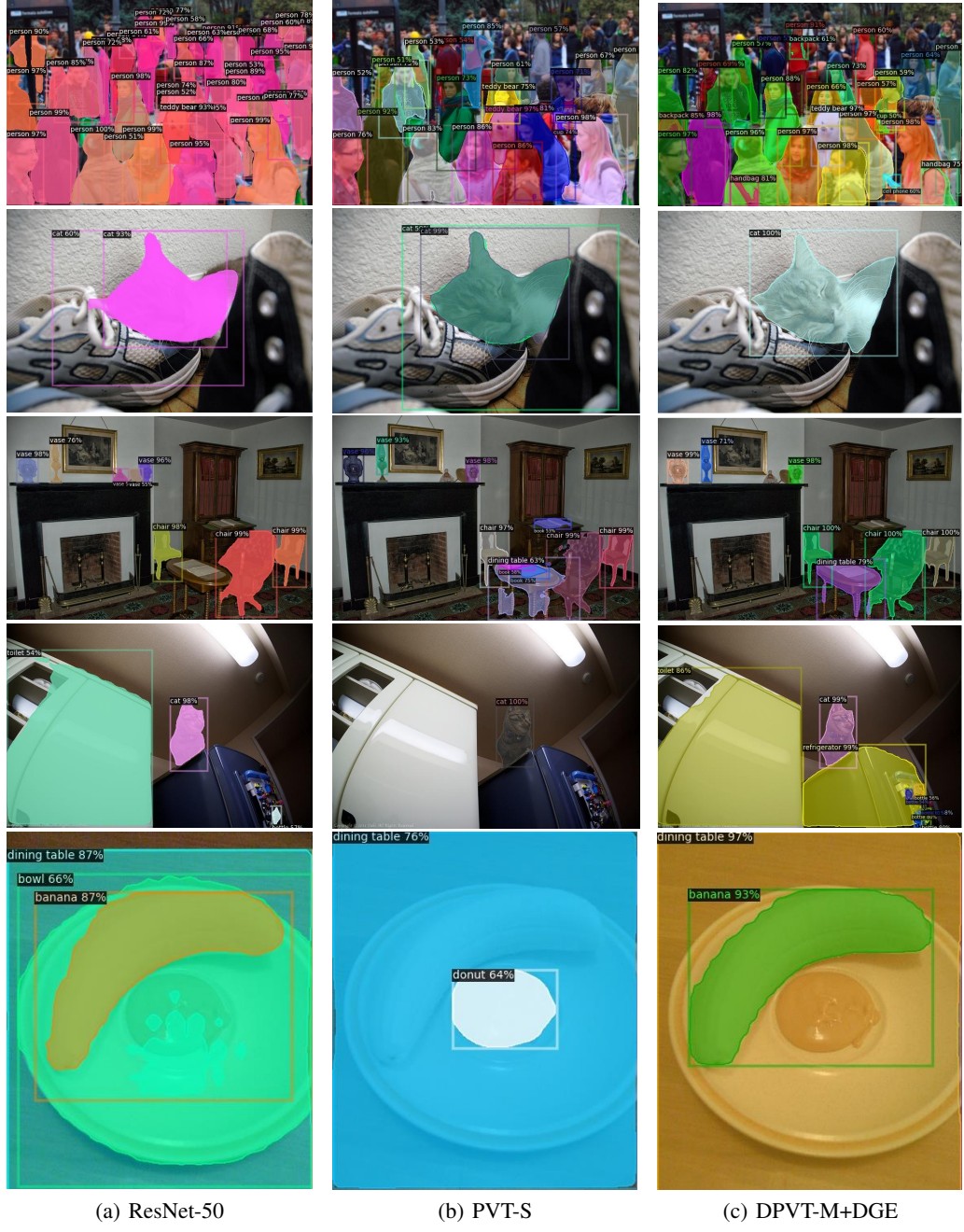

    (a) ResNet-50               (b) PVT-S               (c) DPVT-M+DGE

Figure 7: The visualization of different backbones for object detection and instance segmentation on COCO *val* set. All the models are based on the Mask-RCNN framework and have similar computational complexity.

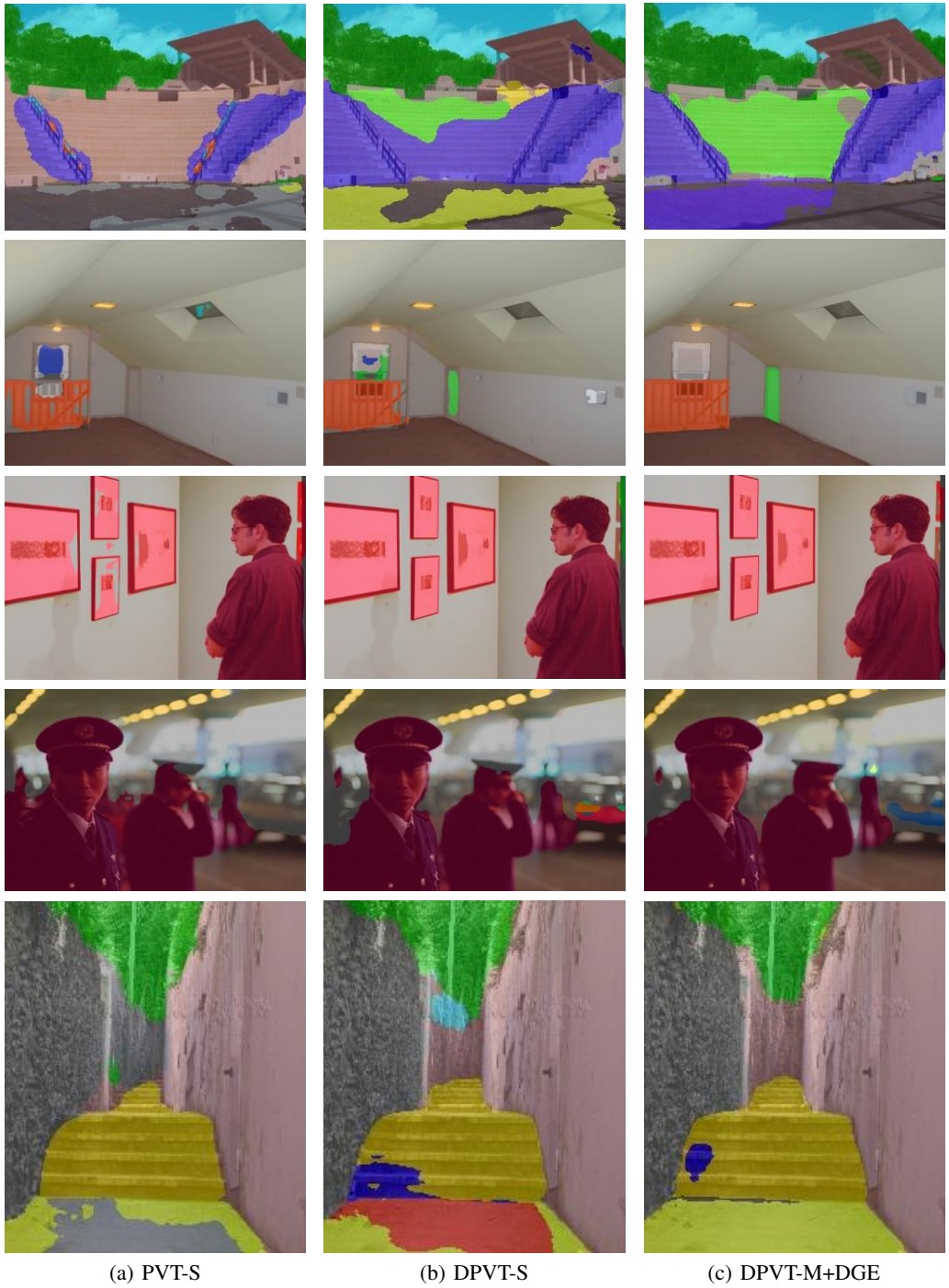

|     |     |     |
| --- | --- | --- |
| (a) PVT-S | (b) DPVT-S | (c) DPVT-M+DGE |

Figure 8: The visualization of different backbones for semantic segmentation on ADE-20K *val* set. All the models are based on the Semantic-FPN framework and have similar computational complexity.