# OpenReview forum: "Dynamic Grained Encoder for Vision Transformers"
_NeurIPS.cc/2021/Conference — NeurIPS 2021 Poster_

### Official Review · Reviewer_6xYd · 2021-07-16

**Rating:** 6
**Confidence:** 5

**Summary:**

The paper proposes a dynamic approach to split an patch into different sizes to save computations for vision transformers. In particular, the authors first split an image into different regions. And for each region, the authors propose a dynamic approach to select one of three predefined patch sizes. This is achieved with a gumbel-softmax function, which enables deriving categorical decisisons in a differentiable way.  The authors experimented with DeiT , PVT  and DPVT.

**Limitations And Societal Impact:**

No.

**Main Review:**

Strengthes:
1. Overall, the paper is well-written and easy to follow.
2. Extensive experiments are conducted with different transformer architectures.
3. The idea of dymamic routing is straightforward.

Weaknesses:
1. Dynamic networks are oftentimes not GPU friendly since they produce instance-specific inputs. It would be nice to discuss the runtime in addition to flops.
2. It would be nice to provide more training details like what is the type of GPUs used and how many hours are used in sec 4.1.1. The authors answered yes to the question "Did you include the total amount of compute and the type of resources use[ type of GPUs, internal cluster, or cloud provided]".  But, these details could not be found in the paper.

**Time Spent Reviewing:**

5

---

> ### Author Response · Authors · 2021-08-10
> **Response to Reviewer 6xYd**
>
> **Q1:** **_It would be nice to discuss the runtime in addition to FLOPs._**
> **A1:** Thanks for the suggestion. We have reported the runtime in Tab.2, Tab.3, and Fig.6 of the appendix. For instance, as shown in Tab.3, the DGE can reduce the runtime by 47.7% over the PVT-S while maintaining comparable performance. Moreover, the PVT-M + DGE achieves 2.1% mIoU absolute gains with less runtime over the PVT-S. Besides, we have discussed the runtime on GPUs and some limitations of our work in the appendix. We will add a figure of the accuracy _w.r.t._ runtime in the final version.
>
> **Q2:** **_It would be nice to provide more training details, e.g., the type of GPUs used and the training time._**
> **A2:** Sorry for the confusion. As presented in Fig.6 of the appendix, we use four compute nodes with 32 Nvidia Tesla V100 GPUs during the training phase. For instance, we spend 1.2 days training the PVT-S + DGE model with 300 epochs. We will clarify it and add more details in the final version.

---

### Official Review · Reviewer_g5J8 · 2021-07-16

**Rating:** 7
**Confidence:** 5

**Summary:**

The authors propose a dynamic grained encoder for vision transformer, which introduces sparse queries to exploit the spatial redundancy. The novel method reduces the computational cost by 40%-60% under comparable performance on image classification. The generalizability of the proposed method is demonstrated on object detection and semantic segmentation. Besides, the authors analyze the intrinsic spatial redundancy of natural images and the properties of the proposed method, which are very interesting.

**Limitations And Societal Impact:**

Yes

**Main Review:**

 Originality:

This work is related to vision transformer and dynamic networks. The authors propose a novel dynamic routing method for vision transformer, which exploits the spatial redundancy and reduces the computational complexity significantly.

Quality:

The submission is technically sound. In Sec. 3.1, the authors analyze the spatial redundancy of natural images, including spatial redundancy statistics, the correlation corresponding to spatial redundancy among different images, and the accuracy impact when reducing the number of queries. It claims the spatial redundancy could be exploited and a data-dependent manner is necessary. The proposed method is evaluated on Image Classification, Object Detection, and Semantic Segmentation.

Clarity:

The submission is clearly written and well-organized.

Significance:

The results are good. The experiments show that the proposed dynamic routing method reduces the computational cost for vision transformer. The generalizability is demonstrated on downstream tasks including object detection and semantic segmentation.

Issues:

1. In Fig. 5, the comparable methods are slightly few on Image Classification. the proposed method does not compare with other dynamic networks like dynamic convolution and MSDNet.

2. The proposed region split and routing method bring extra computational cost. Please compare the actual throughput of all models in Fig. 5 (b) to demonstrate the high efficiency.

3. In Sec. 4.1.1, the input resolution is 256x256, limited to the power of 2. Did you try other input resolutions like 224x224? If yes, how to handle the non-2-power resolution? Could you please provide the performance under 224x224 resolution? (The issue shall be addressed to improve the impact of this work.)

4. Besides data dynamic, it is suggested to consider model dynamic. Autoformer (https://arxiv.org/abs/2107.00651) provides dynamic search of layers for vision transformer, while DVT (https://arxiv.org/pdf/2105.15075.pdf) proposes adaptive sequence length for vision transformer. Both are related to this concurrent work. Pls add them into discussion and comparison.


**Time Spent Reviewing:**

6 hours

---

> ### Author Response · Authors · 2021-08-10
> **Response to Reviewer g5J8**
>
> **Q1:** **_The comparable methods are slightly few on image classification in Fig.5._**
> **A1:** As presented in Lines 90-93 and Lines 234-236, we compare the DGE with the skip-location strategy of dynamic convolutions, and the results demonstrate the superiority of the DGE for vision transformers. We will follow the reviewer's suggestions and update Fig.5 with more comparable methods in the revision.
>
> **Q2:** **_The actual throughput of all models in Fig.5(b)._**
> **A2:** We appreciate the suggestion. As shown in Tab.2, Tab.3, and Fig.6 of the appendix, we report the actual runtime of different models for multiple tasks (object detection, segmentation, and classification). For instance, as shown in Tab.3, the DGE can reduce the runtime by 47.7% over the PVT-S while maintaining comparable performance. Moreover, the PVT-M + DGE achieves 2.1% mIoU absolute gains with less runtime over the PVT-S. We will update Fig.5 with the actual runtime for clarity.
>
> **Q3:** **_Can the DGE handle the non-2-power resolution?_**
> **A3:** Yes. As presented in the footnote of Page 4, the DGE adopts bottom-right padding on the input feature to handle the feature with indivisible resolution. As shown in Tab.8, we provide several experimental results based on $224\times 224$ input images, where the budget $\gamma$ is set to 0.5 for the DGE. Our PVT-M + DGE achieves 1.7% absolute gains over PVT-S with similar computational complexity. We will clarify it and add more results in the final version.
>
> Table 8: Ablation studies of $224\times 224$ input images on ImageNet _val_ set.
>
> | Method | Top1 Acc (%) | Top5 Acc (%) | FLOPs (G) |
> | --- | :---: | :---: | :---: |
> | PVT-S | 79.6 | 95.0 | 3.8 |
> | PVT-S + DGE | 79.4 | 94.6 | 2.1 |
> | PVT-M + DGE | **81.3** | **95.6** | 3.9 |
>
> **Q4:** **_More discussion and comparison with the latest work of dynamic models._**
> **A4:** We thank the reviewer's valuable suggestion! The main idea of DVT is to skip specific spatial locations. It could be seen as a special case of the DGE when the set of candidate granularities $\Phi$ is set to {$0,1$}. We compare the performance in Tab.1 and Lines 234-236. More discussion and comparison will be added to the related work section in revision.

---

> > ### Comment · Reviewer_g5J8 · 2021-09-01
> > **Final comments**
> >
> > I appreciate the authors taking the time, attempting to address the comments through new experiments. The additional experiments address most of my major concerns.
> > After reading other reviewers' comments, I agree with Reviewer 8cdz that there might be a discrepancy between the reported FLOPs and the actual runtime (GPU). It should be carefully addressed in the revision.
> >
> > Overall, I lean to accept this paper and look forward to reading the final version. I've updated my review.

---

> > > ### Author Response · Authors · 2021-09-01
> > > **Response to Reviewer g5J8**
> > >
> > > We greatly appreciate the reviewers' valuable comments. The efficiency of our DGE modules on GPUs mainly relies on the throughput of sparse matrix multiplication, which is dependent on hardware architecture and code optimization. For example, when using the cuSPARSELt library on the latest Nvidia Tesla A100 GPU, the throughput of sparse matrix multiplication can increase by up to 2 times [57]. Besides, the experiments in Tab.2, Tab.3, and Fig.6 demonstrate the reported FLOPs and the actual runtime on CPUs (or GPUs with high-resolution inputs) are almost consistent. This phenomenon indicates that our method has great potential to further improve efficiency by leveraging a sophisticated optimization or using specialized hardware accelerators for sparse operations (e.g., SCNN, Cambricon-X, EIE, and Eyeriss V2). We will clarify it and add a runtime comparison on more devices in the revision.
> > >
> > > [57] Mishra, Asit, et al. "Accelerating sparse deep neural networks." arXiv preprint arXiv:2104.08378 (2021).
> > >
> > > To demonstrate the efficiency of our method, we further optimize the sparse matrix multiplication in the PyTorch framework. The details can be referred to the response titled  "Response to All Reviewers about Throughputs on GPUs".

---

### Official Review · Reviewer_dTZ9 · 2021-07-20

**Rating:** 6
**Confidence:** 5

**Summary:**

This paper investigates the spatial redundancy of natural images and proposes a novel Dynamic Grained Encoder(DGE) for vision transformer. Experiments show that simply replacing the transformer encoder block with DGE can reduce half complexity while the performance is competitive. The visualization shows how it works and the results on object detection and segmentation further verify its feasibility.



**Limitations And Societal Impact:**

Yes

**Main Review:**

- Figure2 clearly illustrates the spatial redundancy. It's not surprising to find the spatial redundancy, but, interestingly, the authors try to divide the patch dynamically. And Figure4 shows it can pay more attention (more queries) for the foreground regions.
- Extensive ablation studies demonstrate the effectiveness of DGE and the authors make convincing conclusions.
- More results on other vision tasks show the generalizability of DGE.

Weakness:

- The definition of P in Figure1 and Figure3 is unclear, is it the $\rho$ in equation(6)?
- "The vanilla encoder block is made up of an MHSA and an MLP", since the length of the token is reduced in GAE, is the residual in MHSA dropped out?
- The authors have considered irregular ways in the paper, like segmentation, and finally simply split images for friendly memory access. However, DGE seems to produce continuous regions for vision transformer. In my opinion, spatial redundancy also exists in discontinuous regions, for example, the wall behind people is similar. Some people design adaptive clustering strategies for queries, like ACT[1], which can be seemed to deal with discontinuous regions. It will be more significant to produce discontinuous dynamic regions.

[1] Zheng M, Gao P, Wang X, et al. End-to-end object detection with adaptive clustering transformer[J]. arXiv preprint arXiv:2011.09315, 2020.



**Time Spent Reviewing:**

10 hours

---

> ### Author Response · Authors · 2021-08-10
> **Response to Reviewer dTZ9**
>
> **Q1:** **_The definition of P in Fig.1 and Fig.3 is unclear._**
> **A1:** The P in Fig.1 and Fig.3 is the $\rho$ in Eq.6. We thank the reviewer's comments and will fix this typo in revision.
>
> **Q2:** **_Is the residual in MHSA dropped out?_**
> **A2:** No. We keep the original structure of the vanilla encoder but only reduce the number of input queries. Besides, as presented in Lines 132-135, we use a refinement procedure to restore the transformed feature to the original resolution while enhancing the details. We will clarify it in the final version.
>
> **Q3:** **_Some suggestions for improving discontinuous dynamic regions._**
> **A3:** We greatly appreciate the reviewer's insightful suggestion. The spatial redundancy in the discontinuous regions is an interesting question well worth studying. In this work, we focus on the inference efficiency of vision transformers, but the irregular ways could bring some adverse effects. We will attempt to compare different splitting ways, _e.g._, ACT, in future work.

---

> > ### Comment · Reviewer_dTZ9 · 2021-09-01
> > **Thanks for your response**
> >
> > I think this is an interesting contribution to vision transformer. I lean to weak acceptance of this paper.

---

### Official Review · Reviewer_8cdz · 2021-07-21

**Rating:** 5
**Confidence:** 2

**Summary:**

This paper introduces sparse queries for vision transformers to exploit the intrinsic spatial redundancy of natural images and save computational costs. Specifically, the authors propose a Dynamic Grained Encoder for vision transformers, which can adaptively assign a suitable number of queries to each spatial region. Concretely, a reshaped 2D feature is first divided into regions using a fixed window. For each region, the number of patches is decided by a data-dependent routing process, and each patch is average pooled to obtain a 1D token. All the tokens are then concatenated into a sequence as the queries. Finally, the output of the encoder is restored to the input resolution by an un-pooling operation and compensates for detailed information with the input feature.

The Dynamic Grained Router is discrete in nature. The authors use the Gumbel softmax trick to train it. To encourage the efficiency, the authors add a (soft) budget regularization to the loss, which can be trained with gradient-based methods.

The authors apply the Dynamic Grained Router to DEIT and PVT, and show that the proposed Dynamic Grained Router can effectively reduce the FLOPs while nearly keeping the accuracy. However, in terms of actual runtime, the overhead from Dynamic Grained Router diminishes the theoretical FLOP gains in the low-resolution region.

**Limitations And Societal Impact:**

Yes

**Main Review:**

There is clearly a discrepancy between the reported theoretical FLOPs and the actual runtime performance. The authors do not present details about how the FLOPs are computed, for a model with a dynamic behavior. The authors may want to elaborate on this.

The visualization in Figure 4 looks interesting. Which layer does this visualization come from? The authors may want to provide per-layer visualization to show the difference cross the early and last few layers. The authors may also want to do an quantitative analysis on ImageNet segmentation (like Table 2 in https://arxiv.org/pdf/2012.09838.pdf), to show how good the dense query region captures the foreground region.

The model for the Dynamic Grained Router (Equation 1) seems very simple, just a linear layer on the pooled feature.  It is surprising that this simple design, optimized with Gumbel softmax, works. Are there any tricks in the training?

As mentioned in Line 225-227, "the deeper layer has lower average computational complexity, which means the deeper layer tends to assign fewer queries. Thus, DeiT is turned into a dynamic feature pyramid structure, which is consistent with the observation in CNNs." This in fact matches the practice in current pyramid transformer arch design (PVT, Vision Longformer, Swin Transformer). Although the authors mention that the Dynamic Grained Router also apply to these pyramid transformers, how much speed gain can it bring with the constraint of no accuracy drop? Based on Figure 5(c), the redundancy is mostly in later layers, whose number of tokens has already been reduced in pyramid transformers. Can the authors give Figure 5(c)-like plot for PVT? Do the authors expect that similar profile will be true for pyramid transformers with local attention mechanism (like Vision Longformer, Swin Transformer)?



**Time Spent Reviewing:**

4

---

> ### Author Response · Authors · 2021-08-10
> **Response to Reviewer 8cdz**
>
> **Q1:** **_There is a discrepancy between the reported FLOPs and the actual runtime._**
> **A1:** We appreciate the comments but believe there exist misunderstandings in efficiency comparison. For instance, as shown in Tab.2, we report the runtime of the proposed models for object detection and instance segmentation.
> It is worth noting that the DGE is only applied to the backbone network other than the entire Mask R-CNN network. As shown in Tab.2, for the Mask-RCNN with PVT-M backbone, the DGE reduces actual runtime by 33.7% and FLOPs by 32.7%. If we only consider the backbone, the DGE (with budget $\gamma=0.5$) reduces the actual runtime by **50.8%**. These results demonstrate that the gains of reported FLOPs and actual speed on CPUs are **almost consistent**.
>
> Moreover, as shown in Fig.6 of the appendix, this conclusion holds for CPUs and the large-resolution inputs on GPUs, which is crucial for mobile devices and many downstream tasks. We believe the optimization on GPUs for small-resolution inputs is capable of being addressed by the community in the future. We will revise the comparison details for clarity in the final version.
>
> **Q2:** **_How the FLOPs are computed?_**
> **A2:** We report the FLOPs following the conventional protocol of dynamic networks [28]. Specifically, we split the entire network into **static** and **dynamic** parts. The complexity of the static part, _i.e._, the modules without dynamic mechanism including the gating networks in DGE, is computed in the standard way [1,3,16]. For the complexity of the dynamic part, _i.e._, the dynamic modules in DGE, we accumulate the complexity associate with each enabled query according to the gating indices. We will clarify it in the final version.
>
> **Q3:** **_Which layer does the visualization in Fig.4 come from?_**
> **A3:** The left and right parts of Fig.4(a) come from stage 1 and stage 2 of PVT, respectively. From left to right, the heatmaps of each instance in Fig.4(b) correspond to stage 1, stage 2, and stage 3, respectively.
>
> **Q4:** **_The quantitative analysis on ImageNet segmentation to show how good the dense query region captures the foreground region._**
> **A4:** We thank the reviewer for the insightful comments and will incorporate this suggestion throughout our revision. Although we focus on a different field from the weakly supervised segmentation [59], we still provide a quantitative analysis to demonstrate the effectiveness of our method.
>
> Following the reviewer's suggestion, we use the metric in [59] to measure the gating scores $\rho$ in each DGE layer. Specifically, we set the candidate granularities $\Phi$ to {$1, 2$}, so that the finer-grained gating scores are taken as a soft-segmentation of the image. We adopt the evaluation protocol in [59] to report the quantitative segmentation results. As shown in Tab.5 and Tab.6, our gating scores have significant superiority even over the weakly supervised method, _i.e._, GradCAM. These results demonstrate that the DGE could guide the transformer to focus on the foreground regions, which is consistent with the visualization. We will incorporate these comparisons in the revision, and all the involved source code will be made public.
>
> Table 5: The quantitative analysis on DeiT-S + DGE ($\gamma=0.5$).
>
> |  | Random | GradCAM [59] | Layer 1 | Layer 4 | Layer 8 |
> | --- | :---: | :---: | :---: | :---: | :---: |
> | Accuracy | 50.0 | 64.4 | 55.4 | 56.3 | **67.6** |
> | mAP | 50.0 | 71.6 | 63.5 | 60.7 | **78.8** |
> | mIoU | 31.9 | 40.8 | 36.4 | 37.7 | **48.2** |
>
> Table 6: The quantitative analysis on PVT-S + DGE ($\gamma=0.5$).
>
> |  | Random | Layer 1 | Layer 6 | Layer 11 | Layer 16 |
> | --- | :---: | :---: | :---: | :---: | :---: |
> | Accuracy | 50.0 | 55.4 | 49.1 | **67.8** | 65.5 |
> | mAP | 50.0 | 68.0 | 45.2 | 71.3 | **79.4** |
> | mIoU | 31.9 | 34.5 | 32.5 | **50.2** | 46.6 |
>
> [59] Chefer, Hila, Shir Gur, and Lior Wolf. "Transformer interpretability beyond attention visualization." Proceedings of the IEEE/CVF Conference on Computer Vision and Pattern Recognition. 2021.
>
> **Q5:** **_The model for the DGE seems very simple, are there any tricks in training?_**
> **A5:** No. For a fair comparison, we strictly follow the training and evaluation protocols in DeiT and PVT except for the image resolution in some experiments (refer to Line 200, we re-train all involved vision transformers). All the source code will be made public.
>
> **Q6:** **_How much speed gain can the DGE bring to the pyramid transformers with no accuracy drop?_**
> **A6:** We have provided the runtime comparison for the pyramid transformers in Tab.2 and Tab.3. For instance, as shown in Tab.3, our DPVT-S + DGE can reduce the runtime by 42% over the DPVT-S with no accuracy drop.
>
> **Q7:** **_Can the authors give Fig.5(c) like the plot for PVT?_**
> **A7:** Yes. We provide the statistics for PVT-S + DGE in Tab.7. We thank the reviewer's suggestion and will add a figure to the final version. As shown in Tab.7, our method can also reduce the spatial redundancy of pyramid transformers, _e.g._, PVT.
>
> Table 7: The computational complexity ratio of different layers in PVT-S + DGE ($\gamma=0.5$ and $\Phi=$ {$1, 2$}), $\sigma$ is standard deviation.
>
> |  | Layer 2 | Layer 6 | Layer 11 | Layer 15 |
> | --- | :---: | :---: | :---: | :---: |
> | Mean | 0.25 | 0.98 | 0.65 | 0.28 |
> | $\sigma$ | 0.00 | 0.05 | 0.11 | 0.07 |
> | Max | 0.25 | 1.00 | 0.94 | 0.63 |
> | Min | 0.25 | 0.70 | 0.31 | 0.25 |
>
> **Q8:** **_Does the DGE work for pyramid transformers with the local attention mechanism?_**
> **A8:** Yes. As presented in Lines 192-195, we propose a new pyramid transformer, _i.e._, DPVT, with deformable attention, which can be seen as a dynamic local-attention mechanism.  As shown in Tab.4, with less computational complexity, our DPVT-M + DGE achieves **4.6%** mIoU absolute gains over the Swin-Ti. We will apply the proposed method to more varieties of vision transformers in the revision.

---

### Author Response · Authors · 2021-08-10
**Response to All Reviewers**

We sincerely thank all the reviewers for their valuable comments and suggestions. We respond to the main concerns of reviewers as follows. We will revise the experimental analysis and improve the writing as suggested by the reviewers. Besides, the source code of all the involved models will be made public to facilitate future research.

---

### Author Response · Authors · 2021-09-02
**Response to All Reviewers about Throughputs on GPUs**

To address the reviewers' concerns about the efficiency on GPUs, we implement an optimized CUDA kernel for batched sparse matrix multiplication. With this kernel, our DGE shows **a significant advantage in throughput on GPUs even with small-resolution inputs**. Specifically, as shown in Tab.9, when inputting a feature with $32\times 32$ resolution, the DGE can increase the throughput on GPUs by about 40%, which is almost consistent with the theoretical computational complexity. The source code of kernel and benchmark based on the PyTorch framework is provided at the anonymous link (refer to the supplementary material). We will update the runtime of all the involved models in the revision.

Table 9: Actual throughput of an encoder layer with DGE ($\gamma=0.5$) on the latest Nvidia RTX 3080Ti GPU. "C" indicates the number of channels.

| Method | FLOPs (G) |  Throughput (item/s) | Throughput Gain (%) |
| --- | :---: | :---: | :---: |
| Encoder (C=384) | 2.4 | 2378.3 | - |
| Encoder + DGE (C=384) | 1.2 | 3731.5 | 36.3 |
| Encoder (C=768) | 8.3 | 1094.3 | - |
| Encoder + DGE (C=768) | 4.2 | 1861.1 | 41.2 |
| Encoder (C=1024) | 14.0 | 744.5 | - |
| Encoder + DGE (C=1024) | 7.1 | 1262.4 | 41.0 |

---

### Decision · Program_Chairs · 2021-09-27

**Decision:**

Accept (Poster)

**Comment:**

The paper introduces a dynamic vision transformer model that reduces spatial redundancy of image features. Three reviewers recommend acceptance, highlighting that the paper is well-written, the idea is interesting, and the experiments are rigorous, demonstrating the effectiveness of the method in different tasks (image classification, object detection, and semantic segmentation). One reviewer considers that the paper does not pass the acceptance threshold, due to unclear actual runtime gains when images have low-resolution, a concern shared by other reviewers as well. In the rebuttal, the authors demonstrated that a significant advantage in throughput on GPUs can be obtained through their implementation of  an optimized CUDA kernel for batched sparse matrix multiplication. The AC finds this response convincing, and agrees with the majority that the paper passes the acceptance bar of NeurIPS. The authors are encouraged to add the discussion in the rebuttal to the final version of the paper.